# Application of Gelatin Composite Coating in Pork Quality Preservation during Storage and Mechanism of Gelatin Composite Coating on Pork Flavor

**DOI:** 10.3390/gels8010021

**Published:** 2021-12-27

**Authors:** Haoxin Li, Renrun Tang, Wan Aida Wan Mustapha, Jia Liu, K. M. Faridul Hasan, Xin Li, Mingzheng Huang

**Affiliations:** 1The Key Laboratory of Environmental Pollution Monitoring and Disease Control, Ministry of Education, School of Public Health, Guizhou Medical University, Guiyang 550025, China; lhx@gmc.edu.cn (H.L.); T15885904106@163.com (R.T.); 2Department of Food Sciences, Faculty of Science and Technology, Universiti Kebangsaan Malaysia (UKM), Bangi 43600, Malaysia; wanaidawm@ukm.edu.my; 3Institute of Food Processing Technology, Guizhou Academy of Agricultural Sciences, Guiyang 550006, China; mcgrady456@163.com; 4Simonyi Karoly Faculty of Engineering, University of Sopron, 9400 Sopron, Hungary; faridulwtu@outlook.com; 5College of Food and Pharmaceutical Engineering, Guizhou Institute of Technology, Guiyang 550025, China

**Keywords:** gelatin composite coating, coating preservation, pork flavor, metabolic mechanism

## Abstract

Gelatin coating is an effective way to prolong the shelf life of meat products. Aiming at solving the problem of flavor deterioration during the storage of pork at room temperature, pork coating technology was developed to preserve the pork at 25 °C, and the comprehensive sensory analysis of vision, touch, smell, and taste was used to study the effect of coating on preservation of pork flavor. Herein, uncoated (control) and coated pork samples (including gelatin coating and gelatin coating incorporated with ginger essential oil) were analyzed to investigate the integrity of pork periodically during storage at 25 °C for weight loss, color, texture (springiness, chewiness, cohesiveness, gumminess, and hardness), microstructure, odor (electronic nose), taste (electronic tongue), volatile flavor substance, and taste ingredients. The results suggested that ginger essential oil (GEO) gelatin coating and gelatin coating can effectively inhibit the loss of water dispersion and slow down the oxidation reaction, coating treatments could significantly (*p* < 0.05) retarded the weight loss of pork slices, with values of 20.19%, 15.95%, 13.12% for uncoated, gelatin coated, and GEO-gelatin coated samples during 24 h of storage, respectively. Compared with control group, the color, texture, smell, and taste evaluations demonstrated that coating treatments had improved sensory and texture attributes during the storage period. Furthermore, the comprehensive results from the physical property assays (especially the texture), morphological assay and volatile odor assays showed that the GEO-fish gelatin composite coating had better preservation effect on pork flavor than the fish gelatin coating. The study suggests that the gelatin composite coating could be developed as a prospective active packaging to preserve pork meat at room temperature.

## 1. Introduction

At present, China is the world’s largest meat producer, with an annual output of 41.4 million tons, with pork accounting for 2/3 among the whole output. At the same time, the production and consumption of pork worldwide are also steadily increasing [1].

Domestic pork is sold from producers to agents and retailers. The selling venues are mainly focused on supermarkets and farmer’s markets generally at room temperature. In fact, fresh pork is often wasted due to a short shelf life during storage, simple plastic packaging is commonly used in the sales process, which leads to accelerated pork spoilage and deterioration of flavor and appearance. Thus it is impossible to meet the consumers’ demand for high-quality pork. The huge consumer market is in urgent need for efficient, safe, and environmentally friendly ways to keep pork fresh. Therefore, the preservation of meat and meat products has received widespread attention in recent years.

Edible coating refers to a thin primary packaging layer prepared from biological materials of food origin. The edible coating can form a solidified coating on the food surface, and the barrier effect of the coating can keep the food fresh [2]. Hydrocolloids and lipids derived from polysaccharides and proteins are the main components of edible coatings [3]. The application of edible coatings can be an alternative technology to extend the shelf life of meat because they provide a barrier to prevent oxygen permeation, water transfer, drip loss, lipid oxidation, microbial growth, and act as food additives carriers, such as antioxidants and antibacterial agents, etc. [4]. Gelatin is an edible coating material derived from protein. Gelatin possesses good coating-forming properties, degradability, transparency, and biocompatibility. Adding bioactive additives (such as essential oils and natural product extracts) to gelatin can enhance the antibacterial and antioxidant properties of the coating [5].

The effects of composite coating on pork quality were studied by characterizing physical and chemical properties (Color, pH, TBars, Thiol Group) and microbiological properties (TVC). It was found that the composite prepared by chitosan-gelatin-grape seed extract had the best preservation effect on pork, and could effectively prolong the shelf life of pork. The effects of chitosan-gelatin coatings containing tarragon essential oil (TEO) or TEO-loaded nanoparticles (TEO-NPs) on the preservation of pork slices during refrigerated storage for 16 days were studied. Recently it was indicated that nano-encapsulation contributed to the sustained release of TEO and caused an improved antioxidant, antibacterial and sensory properties [6]. However, it is worth mentioning that most of the past studies focused on the detection of pork physical and chemical indexes and microbial indexes, and there were very few reports focused on the volatile flavor and taste of pork. Flavor, as a key index of meat quality, is closely linked to consumers’ intuitive feelings. The research on meat flavor can better reflect the preservation effect of coating on pork. According to Spence (2015), food flavor is not experienced by a single sense, but by multiple senses, such as vision, hearing, touch, smell, and taste [7].

Essential oil may be an effective way to retard the deterioration of meat product, it is a natural extract with antioxidant and antibacterial properties [8]. Adding essential oil to the preparation of gelatin coating can enhance the biological activity of the gelatin coating [9], and their incorporation into active coating is an alternative to synthetic additives currently utilized in food product. However, there were few reports of GEO applications in coating formulation for meat preservation. The objective of this work was to employ gelatin as the raw material and ginger essential oil as the additive to prepare a composite coating for the preservation of pork. Weight loss, color, and microstructure, texture, smell (volatile flavor detection and electronic nose detection), and taste (electronic tongue detection) assays were carried out to explore the influence mechanism of coating preservation technology on pork flavor during storage at 25 °C.

## 2. Results and Discussion

### 2.1. Physical Properties

#### 2.1.1. Weight Loss

The weight loss rate of pork samples with different coating at 25 °C storage temperature is shown in Figure 1. The weight loss of control group indicated the highest value (0–20.19%), higher than the weight loss of gelatin coating (0–15.95%) and GEO-gelatin coating (0–13.12%). The weight loss rate of the coated group was lower than that of the control group, and the effect of the ginger essential oil gelatin group was slightly lower than that of the gelatin group (*p* < 0.05). As regards the moisture distribution, it was reported that muscle tissue contains 75% water content among which only 10 to 15% was bound water. Muscle tissue had a relatively higher proportion of bound water and more liable to lose, thus pork can easily lose water during storage [10]. Water content of meat is a critical parameter of meat tenderness. A low weight loss rate indicates that the pork has higher moisture content, and higher moisture content in the pork indicates a relatively better tenderness of the meat. The coating acts as a water vapor barrier to effectively prevent the weight loss of pork; on the other hand, gelatin is regarded as a hydrophilic colloid [5], and water vapor is easily absorbed and permeated on the coating. Adding the hydrophobic ingredient ginger essential oil can reduce permeability property of the coating to water vapor, hence the weight loss rate of the essential oil gelatin group was lower than that of the gelatin coating group.

#### 2.1.2. Color

The influence of coating preservation on pork color is shown in Figure 2. The change in the color of pork was expressed by parameters including Δ*E*, *L**, *a**, and *b**. *L** value represents brightness darkness, a value represented red/blue, b value represents yellow/green, Δ*E* value represents total color difference. The results indicated that with the prolonged preservation period, *a** value, *b** value, and ΔE value increased at room temperature, while *L** value decreased. Compared with the control pork sample, the changes in the color of two different coating pork samples were relatively slow. In details, GEO-gelatin coating pork samples performed better than gelatin coating pork sample. The degradation of esters, pigments, proteins, carbohydrates, and vitamins produces oxidative products, furthermore these productions result in color changes [11]. Enzymatic hydrolysis of carbohydrates, fats, and proteins give rise to the meat turning into green color [12]. Therefore, with the addition of storage time, pork would gradually lose the initial color and turn into green. The moisture loss of pork led to the decreased brightness. Without the coating protection, the control sample was liable to lose moisture and further decreased in brightness [13]. During storage, the moisture inside the pork seeps out, and the gelatin coating possesses a certain water vapor transmission, thus moisture gradually dissipates as it seeps out. Woodmansee and Abbott (1958) reported that Myvacet-coated broiler legs had a weight loss after storage at 4 °C for 10 days, due to dehydration of 4.2 to 6.3% as opposed to a weight loss of 15.1 to 30.2% for uncoated control broiler legs. In addition, coated broiler legs revealed less skin darkening during storage than uncoated control samples [14]. This result was similar to our study, the weight loss rate of coated samples were much lower than uncoated samples, the *L** value decreased for both coated and uncoated control samples. The *L** value in meat generally varies with the structural characteristics of the muscle, the water distribution in the muscle, and the position of the muscle. The decrease in *L** value is due to the gradual decrease in the moisture content of the pork, which increases the concentration of myoglobin and other pigments in the muscle towel and reduces the transmission of light. With respect to *a** value, the content of myoglobin is the main factor affecting the color of pork, and it also directly influences the consumption and the acceptability of the consumer [15]. With the extension of storage time, the *a** value of all treatment groups decreased first and then increased, In the early storage period, deoxymyoglobin was dominant, which made the pork meat appear dark red, and the *a** value of all test groups decreased. Subsequently, the residual oxygen in the package reacted with myoglobin producing unstable oxymyoglobin, *a** value rose again after a fall, Compared with other two groups, the *a** values of GEO-gelatin groups maintained a slightly higher value (*p* < 0.05). Kroll et al. (2001) found that phenolic substances, which are regarded as the main ingredient in ginger oil, can directly react with myoglobin and delay the oxidation and discoloration of myoglobin [16]. In addition, the coating acted as a barrier between the pork samples and the oxygen in the storage environment, coating preservation can reduce the color change of pork due to oxidation reaction, considering GEO has antioxidant and antibacterial effect, GEO-gelatin coating can prevent the pork samples from color change arising from the enzymatic hydrolysis and microbial metabolism. Therefore, GEO-gelatin coating preservation on pork could prevent the color change of pork effectively at room temperature.

#### 2.1.3. Texture Analysis

Meat texture and palatability are indispensable attributes influencing consumers’ choice. The influence of coating preservation on pork texture is shown in Figure 3. Hardness, springiness, cohesiveness, gumminess, chewiness are some important parameters of texture characteristics of meat [17]. Hardness (N/cm^2^), Springiness (cm), Cohesiveness (A_2_/A_1_) are 3 independent variables in texture characteristics. Although gumminess and chewiness are dependent variable on the basis of 3 independent variables. The formulas are gumminess (N/cm^2^) = hardness × cohesiveness, chewiness(N/cm) = hardness × springiness × cohesiveness respectively [18]. At the room temperature, with the addition of storage time, the parameters including hardness and chewiness of pork control were increased, while the parameters including springiness, gumminess and cohesiveness were decreased with the prolonged storage time. Both coating and control pork presented opposite trends on parameters of chewiness. Coating preservation could alleviate the variable trend of parameters of pork texture. GEO-gelatin coating performed better than gelatin coating on alleviating the parameter change.

The main factors of texture change of pork during storage are related to the result of the synergy of water content, microbial metabolism and enzyme autolysis. The growth of meat microorganisms will degrade the tissue structure of meat and form meat mud, thus reduce the water holding capacity of pork [12]. The enzyme autolysis of protein, carbohydrate, and lipid give rise to the meat turning to tender in texture [19]. However, with the extension of storage time, the reason for the increase in pork hardness was mainly due to the rapid loss of water on the surface of the pork. The coating acted as a water vapor barrier to suppress the loss of water vapor, and the inhibitive effect of the coating with ginger essential oils was slightly better than that of the coating without ginger essential oils. Based on these two reasons, the hardness of pork would gradually slightly increase during storage. The disintegration and division of myofibrils will lead to a decrease in the cohesiveness value of pork [20]. In terms of another texture parameter-springiness, the elasticity of the muscle can reflect the binding state to the muscle tissue of the pork body. The better the binding ability, the higher was the elasticity. Both springiness and chewiness reflect the edible taste of pork muscle [18]. At the same time, the gumminess and chewiness of pork would change with the variance of harness, springiness, and cohesiveness. On the condition of 25 °C, 35% relative moisture, the hardness of pork would significantly increase, owing to the relatively high temperature and relatively low moisture that led to the surface of pork lose in moisture rapidly and consequently became harder at the surface of the pork. On the one hand, the chewiness of the control group fluctuated significantly, indicating that the quality of uncoated pork had changed greatly and the taste was poor. On the other hand, as a water vapor barrier, the coating can maintain the texture of pork. The antioxidant and antibacterial functions of GEO can prevent the effect of microbial metabolism and auto enzymatic hydrolysis on the texture of pork, thus pork manifested better texture characteristics during storage. Generally speaking, coating can maintain the texture characteristics of pork, compared with the gelatin coating, GEO-gelatin coating performed better on maintaining the texture characteristics of pork.

### 2.2. Morphological Analysis

The effect of different coating on the microstructural properties of pork meat samples was observed using SEM. As shown in Figure 4, the microscopic morphology of pork between the control group and 2 different coating groups was distinguishably different. It was found that under the condition of 25 °C, SEM images displayed the rough and wrinkled structure of control group, Although pork surfaces were smooth in both coated groups, this can be explained by the coating treatment protecting the evaporation on the surface and so, less shrinkage is observed compared to the control group (non-coated) during storage. The difference in the microstructure of the pork meat can be employed to explain the difference in the texture of the pork meat. When the pork meat was stored at 25 °C, due to the evaporation of water on the inner surface, the high temperature damaged the open structure and intercellular spaces of the pork meat. Particularly, the water content of pork meat is tightly related to its brightness, textural hardness, and chewiness. For the two coating groups, less shrinkage was observed in the samples as compared with the control group. Aksoy et al. (2019) studied that the less damage to the porosity and open-pore structure of meat during processing, the meat quality was better [21]. The pork meat of the coating group manifested a better texture than the pork meat of the control group. Furthermore, the pork meat of the coating presented a smooth surface owing to the coating covered the pores on the surface of the pork meat, preventing the water from evaporating and the formation of wrinkles, which indicated that the coating had a protective effect on pork and prevented the wrinkling performance of pork meat due to water loss during storage.

### 2.3. Volatile Odor Analysis

#### 2.3.1. GC-MS

Volatile flavor substances in pork meat are one of the significant indicators for evaluating meat quality. The changes of volatile substances in pork samples with different treatment are shown in Table 1. Categories including 31 kinds of volatile flavor compounds were detected in this experiment. The identified VCs were: alkane (12), alkene (2), alcohol (6), aldehydes (7), and acids (4). These flavor substances are mainly decomposed lipolysis and lipid oxidation, proteolysis, decomposition of carbohydrate such as Strecker degradation, Maillard reaction, and other pathways [22].

Similar to many other studies on meat flavor, aldehydes indicated low odor threshold and might play a significant role in the flavor of meat [23]. Most aldehydes were derived from lipid oxidation: decanal, nonanal, octanal, pentanal, hexanal, and benzenepropanal. Hexanal along with other aldehydes contribute positively to meat flavor, but may generate undesirable flavors at higher concentrations. Dodecanal is associated with waxy and soapy smell, while octanal has green and fruit like flavor. Unsaturated aldehydes are responsible for the fat aroma of meat and play some part in species-characteristic flavor, such as the 8–9 carbons n-2-alkenals [24].

The sum of aldehydes in fresh pork meat, control group, gelatin-coated pork meat and GEO-gelatin coated pork meat are 21.63 × 10^−3^ μg/μL, 216.74 × 10^−3^ μg/μL, 63.42 × 10^−3^ μg/μL, and 43.69 × 10^−3^ μg/μL, respectively. It can be illustrated that the total amount of aldehydes in pork meat increases with the extension of storage time. The coating can alleviate the increasing trend of the total aldehydes, particularly the GEO-gelatin coating performs better than that of gelatin coating in this respect. 

Compared with aldehydes, alcohols have higher odor thresholds and thought of as minor flavor contributors to meat products. However, as a precursor of aldehydes and ketones, it has additive effect on flavor formation. The main flavor attributing factors including 3-Phenylpropanol which has a sweet fruit-like fragrance, 2-Propen-1-ol, 3-phenyl- presents a cream fragrant odor. Control pork samples contained higher amounts of 1-Penten-3-ol than coating treatment groups, those alcohols were typical markers of raw meat [23]. 

Acidic compounds mainly arise from small molecular fatty acids produced during the hydrolysis and oxidation of fat. Generally, fresh pork had higher content than other groups’ content of acidic compounds. The acids have relatively high odor threshold, and only 4 types of acids were detected, which had little effect on the whole flavor of pork meat.

Finally, coating treatment groups was characterized by the presence of three terpenes that seemed unrelated to the typical meat flavor, however it could result from the specific compounds in coating material. 

Figure 5 is a clustering heat map of the volatile odor of pork samples with different processing methods. The composition and content of volatile odors in each pork sample is expressed by different colors in the heat map. It can be seen (Figure 5) that the gelatin coating, GEO-gelatin coating and fresh pork are in one sub-category, indicating that the odor of coated pork is closest to that of fresh pork. On the other side, the control group of pork is in the other sub-category, furthermore, with respect to the fresh pork, compared with other groups, the distance between the fresh pork and control group was the farthest, that is to say the volatile odor of pork between gelatin-coated pork and control group was the most distinguished difference from that of fresh pork.

#### 2.3.2. E-Nose Radar Image Analysis and PCA

In order to study how the two different coatings affect the final sensory quality of pork samples, typical response of the sensor array toward different treatment of pork samples is presented (Figure 6A). Fresh pork samples were used to evaluate the freshness of the pork samples. It can be observed that, there exited slight difference between fresh pork and 3 pork samples with different treatment towards the S1, S2, S3, S5, and S10 at 25 °C. On the contrary, referring to S4 and S6, it revealed relatively distinguished difference compared with fresh pork. On the whole, S1, S2, S3, S5, S10 were not sensitive to pork odors. Coating preservation on pork could alleviate the odor change of pork effectively at room temperature. Furthermore, the data obtained from E-nose also confirmed that compared with the gelatin coating, adding ginger essential oil to gelatin-based coating was beneficial in maintaining flavor quality of pork. 

In this work, the data were inspected by PCA in order to reduce the large set of variables and obtain a small number of linear combinations, during the inspection process, the response of sensor S4 and S6 increased sharply in the initial period, reached to the peak values at 6–16 s, and finally reached to a stable equilibrium. The response of S1, S3, and S5 increased slowly with time elapsing and stabilized after 70 s. As for the other sensors, there were no obvious changes in G/G0 values during measurement time. The sensors signals used for multivariate analysis were generally stabilized. Therefore, the mean values of responses from 75 s to 78 s for each sensor were calculated as an original dataset. On the other hand, the PCA analysis of the E-nose can reflect the changes of volatile components in pork samples. As shown in Figure 6B, the summed sample variance of 87.94% could be observed from PC1 (68.56%) and PC2 (19.38%) which exceeds 85%. Therefore, samples from different treatments were distinguishable from each other by PCA. It can be observed from the figure that the 25 °C GEO-gelatin coated pork was the closest to the fresh pork sample, followed by the 25 °C gelatin coated pork, finally the 25 °C control group. Compared with the control pork, the distance between the coated group and the fresh pork was closer, and the effect of GEO-gelatin coating was better than that of gelatin coating. 

Oxidation, microbial growth, and self-enzymatic hydrolysis are the three basic mechanisms of meat spoilage during processing and storage. Lipid oxidation and microbial growth can cause changes in the odor of meat [12]. Fat oxidation exerted a greater impact on meat flavor. Despite the proportion of lipid oxidation is small; the flavor changes are still significant [25]. From the PCA diagram, the following conclusions can be drawn: the volatile smell of pork stored at 25 °C was closer to that of fresh pork; compared with the control group, the coating can maintain the fresh odor of pork better, the effect of GEO-gelatin coating was better than that of pork gelatin coating.

#### 2.3.3. E-Tongue Radar Image Analysis and PCA

Typical response of the taste array toward different treatment of pork samples is presented (Figure 7A), the radar image can be used to represent the response values of the eight sensors of the electronic tongue to the taste of pork samples. As shown in Figure 7A, there was no obvious difference between the three groups of pork and fresh pork under the condition of 25 °C in the saltiness sensor, aftertaste-B and the astringency sensor. With the extension of storage time, the changes in the taste of pork meat gave rise to the increased acidity, decreased bitterness, decreased aftertaste-B, and increased richness. In general, coating can maintain the taste of pork, compared with the gelatin coating, GEO-gelatin coating performed better on maintaining the taste characteristics of pork.

PCA response value of E-tongue in pork samples is presented in Figure 7B, the PCA analysis of the E-tongue can reflect the changes of taste in pork samples. As shown in Figure 7B, the results of PCA demonstrated that the contribution rate of the first axis was 80.14%, the contribution rate of the second axis was 11.58%, and the cumulative contribution rate of the two axes was 91.72%. It is shown that the principal component analysis contains majority of the information in the electronic tongue data. It can be observed from the figure that the 25 °C GEO-gelatin coated pork was the closest to fresh pork sample, followed by the 25 °C gelatin coated pork, finally the 25 °C control group. Compared with the control pork, distance between the coated group and the fresh pork was closer, and the effect of GEO-gelatin coating was better than that of gelatin coating. The changes in pork taste were related to the self-decomposition of pork muscles and the growth of microorganisms. Given the fact that during storage, changes in the composition and content of free amino acids affects the taste and flavor of pork. The coating can act as an oxygen barrier, reducing the contact between pork muscle and air, thereby slowing down the change of free amino acids caused by muscle self-decomposition and microbial growth. GEO possesses antioxidant and antibacterial activity, which can effectively prevent the self-decomposition of pork muscle and inhibit the growth of microorganisms. Hence, GEO coating had a better effect on inhibiting the metabolism of free amino acids.

## 3. Conclusions

Based on the results in this work, compared with the control group, the weight loss of the coated pork was relatively low; coating can effectively decrease the weight loss of pork, meanwhile, the color and texture characteristics of the pork can be maintained. Furthermore, GEO-gelatin coating performed better on maintaining the color and texture characteristics of pork. According to the results obtained from GC-MS, this coating preservation method can alleviate the accumulation of volatile components in pork, thereby maintaining the volatile flavor of pork. On the other hand, coating preservation on pork could alleviate the odor and taste change of pork effectively at room temperature. Furthermore, the data obtained from E-nose and E-tongue also confirmed that compared with the gelatin coating, adding ginger essential oil to gelatin–based coating was beneficial in maintaining flavor and taste quality of pork. In light of comprehensive analysis of physical properties and changes of volatile flavor compounds of pork meat at different treatment methods, the GEO gelatin composite coating had the best effect on the preservation of pork flavor. This work presented a new endeavor and contribution toward the effect of gelatin composite coating on pork flavor metabolism mechanism. Furthermore, work could be undertaken to explore the effects of gelatin composite coating on the nutrients and safety of pork meat.

## 4. Materials and Methods

### 4.1. Materials

Fifteen fresh pork fillets were purchased from a local supermarket in Yonghui supermarket (Guiyang, China) at 24 h post-mortem. Fish gelatin with gel strength 270 degree was kindly supplied by Jiliding Biotech Co., Ltd. (Jiangsu, China). Tween 80 and glycerol were obtained from Sigma-Aldrich Co., Ltd., (New York, NY, USA) and ginger essential oil (GEO) was provided by Yumei Cosmetic Co., Ltd. (Jiangxi, China). All reagents were analytical graded and commercially available. Deionized water was used for conducting all the experiments.

### 4.2. Coating Preparation

At first, gelatin (8%, based on deionized water) particles were melted in deionized water, and then glycerin (10%, *w*/*w*, based on the weight of gelatin) was added to the solution with continuous agitating for 10 min to obtain a uniform solution. This is liquid A; the content of liquid B was used to prepare mass ratio of Tween 80 and GEO mixtures 1:1. Active GEO-gelatin coating solution was prepared by adding liquid B to liquid A (GEO with a final concentration of 0.5% *v*/*v*). With respect to gelatin coating, this solution was prepared according to the preparation of GEO gelatin coating without adding GEO. All the film-forming solutions were prepared and utilized simultaneously.

### 4.3. Pork Sample Preparation

In a sterile environment, fifteen fresh pork boneless fillet were irradiated with a UV lamp on each side at a distance of 10–15 cm for 30 min prior to cutting into cuboids of dimensions 4 cm × 3 cm × 2 cm using a pre-sterilized knife and cutting board, subsequently the pork samples were prepared and placed in sterilized petri dishes. All reagents and operating apparatus in contact with pork samples must be sterilized at 121 °C, under high pressure for 15–20 min in advance. The cuboids were immersed into two types of coating-forming solutions, respectively, at room temperature, and then promptly stored for 1 min at-20 °C, at relatively low temperature, the coating-forming solution could be capable of forming a solid coating at the surface of pork cuboids rapidly. The meat was wrapped using fish gelatin and GEO-fish gelatin coating and casted on a square steel plate (150 mm × 150 mm), then stored in a constant temperature incubator at 25 °C for 0, 3, 6, 9, 12, 15, 18, 21, 24 h, respectively. At the specific time, three parallel meat samples for each coating was taken out and cut in a sterile environment for further analysis during 24 h of storage. Consequently, the meat slices were randomly divided into three groups: (i) Uncoated (Control); (ii) coated with fish gelatin; (iii) coated with GEO-fish gelatin. Fresh pork was prepared as blank sample for SEM test, volatile components test, e-nose test, and e-tongue test. The procedure of pork from purchase to coating process should be accomplished within 1 h [26].

### 4.4. Physical Property

#### 4.4.1. Weight Loss

The pork weight loss rate was expressed according to the Formula (1).
(1)$$\mathrm{Weight}\;\mathrm{loss}\;(\%)=\frac{\left(M_{1}-M_{2}\right)}{M_{1}}\;\times \;100\%$$
where *M*_1_ is the initial weight of the pork and *M*_2_ represents the final weight of the pork. All tests were carried out in triplicate [27].

#### 4.4.2. Color Analysis

The color of pork was measured by colorimeter [28] (Konica Minolta Optics, Tokyo, Japan). The total color difference was expressed according to the Formula (2).
(2)$$∆E=\sqrt{{\left(L-L^{*}\right)}^{2}+{\left(a-a^{*}\right)}^{2}+{\left(b-b^{*}\right)}^{2}}$$
where *L* is the initial light and dark color of the pork, where *a* is the initial red-green color of the pork, and where *b* is the initial yellow-blue color of the pork. *L** represents the final light and dark color of the pork and *a**** represents final red-green color of the pork. *b**** represents final yellow-blue color of the pork. All tests were carried out in six times [29].

#### 4.4.3. Texture Profile Analysis (TPA)

The texture characteristics (springiness, chewiness, cohesiveness, hardness, and gumminess) of pork were determined using a Texture Analyzer (TA-XT Plus, Stable Micro Systems, Surrey, UK) assembled with a 36 mm (P/36R) diameter cylindrical probe in the texture profile analysis (TPA) test mode. The samples were sliced into pieces before the test. The main conditions of the double compression test were as follows: pre-test speed 1 mm/s, test speed 1 mm/s, post-test speed 5 mm/s, trigger force 5 g, 5 s gap between compressions and compressed to 50% of its height. For each type of pork, samples from three independent batches were tested with more than eight times [30].

### 4.5. Scanning Electron Microscopy Investigation

For scanning electron microscopy (SEM) (S-3400N, Hitachi, Japan), the pork samples were sliced into thin pieces then freeze-dried. The SEM test was exploited to analyze the microstructure of freeze-drying pork samples. Samples were mounted on copper stubs and sputter-coated with gold. Samples were then examined and images were recorded with a SEM machine at accelerating voltages of 20 kV and viewed at magnification levels of 2000×. 

### 4.6. Volatile Odor Analysis

#### 4.6.1. Gas Chromatography-Mass Spectrometry

Firstly, simultaneous distillation extraction (SDE) method was used to extract the volatile smell of pork meat. Briefly, pork (100 g) plus 500 mL deionized water were added to a 1 L round-buttoned flask connected to a Likens-Nickerson type SDE apparatus. The volatiles were extracted with 250 mL dichloromethane for 2 h in slight boiling state. The volatile odor in pork sample was transferred from the water phase to the dichloromethane phase. The dichloromethane was then concentrated to approximately 2 mL by Wechsler distillation column. Previously, an appropriate concentration of 1, 2-dichlorobenzene has been added to the concentrate as internal standard to eliminate errors caused by volume changes. All samples were combined and concentrated to 2 mL with a Vigreux column before subjected to GC–MS analysis Subsequently concentrates were analyzed by a gas chromatograph-mass spectrometer (Trace 1300/ISQ, Thermo Fisher Ltd., USA), equipped with a split/splitless injector and with a HP-5MS 30 m × 0.25 mm × 0.25 μm fused silica column from Sigma-Aldrich (using a helium flow rate of 1 mL min^−1^). The column temperature was initially held at 50 °C for 2 min, gradually increased to 160 °C at a rate of 3 °C/min, then to 220 °C at a rate of 5 °C/min. The mass spectrometer working conditions were as follows: electron energy was set to 70 eV; using helium as a carrier gas, the constant flow rate was 1 mL/min, sample injection volume was 1 μL. All tests were carried out in triplicate [31].

#### 4.6.2. The Electronic Nose (E-Nose)

The electronic nose analyzer (Airsense Corporation, Germany, PEN3) was used to obtain the chemical components of the pork samples. The sensor array was composed of 10 different metal oxide sensors (MOS) positioned into a small chamber. Each sensor has a certain degree of affinity towards specific chemical or volatile compounds, and the nomenclature and characteristics of the sensors are designed as follows: W1C (S1), sensitive to aromatics; W5S (S2), sensitive to nitrogen oxides; W3C (S3), sensitive to ammonia, aromatic molecules; W6S (S4), sensitive to hydrogen; W5C (S5), sensitive to methane, propane, and aliphatic non-polar molecules; W1S (S6), sensitive to methane; W1W (S7), sensitive to sulfur-containing organics; W2S (S8), sensitive to broad alcohols; W2W (S9), sensitive to aromatics, sulfur- and chlorine-containing organics; W3S (S10), sensitive to methane and aliphatic. The experimental conditions for E-nose were given as follows: 15 g of the minced pork was placed in a double-layer sealed plastic glass at room temperature, and the beaker was sealed by double-layer plastic for a headspace generation time of 15 min. The headspace generation was carried out to increase the volatile compounds from the pork sample. Before one sample was detected by E-nose, the sensors were cleaned with the flow of fresh dry air, so that the sample can be tested [24]. Thereafter, the sensors were exposed to sample volatiles and the changes in sensors’ responses were acquired by the data acquisition system. During the sampling process, the sample gas was transferred into the sensor chamber at a flow rate of 400 mL/min, the detecting time was 100 s and the cleaning time was 100 s. The multidimensional signals of the E-nose required some data pretreatment before the statistical analysis was performed via radar images. Feature extraction and selection was completed by following the similar method of Principal component analysis [32]. The E-nose measurement was performed at 25 ± 2 °C. All tests were carried out in triplicate.

#### 4.6.3. Electronic Tongue (E-Tongue)

The electronic tongue analyzer (Insent Ltd., Tokyo, Japan TS-5000Z) was used to analyze the taste and flavor of the pork samples. the characteristics of the sensors were designed as follows: Sourness, Aftertaste-A, Aftertaste-B, Umami, Richness, Saltiness, Astringency, and Bitterness. Briefly, 50 g pork sample was stirred with a food processor for 1 min, 200 mL 40 °C deionized water was added and mixed for 1 min before the mixture was centrifuged at 3000 rpm/min for 10 min. Then, 35 mL supernatant extracted was immediately analyzed with the E-tongue. The testing temperature was 20 °C [33]. The E-tongue was composed of eight different receptor containing eight different taste sensors. The multidimensional signals of the E-nose required some data pretreatment before statistical analysis was performed via radar images. Feature extraction and selection was done by method of Principal component analysis (PCA). All tests were carried out in triplicate.

### 4.7. Statistical Analysis

All statistical analyses were performed using Origin Pro 2018(Origin Lab, USA) and SPSS 25.0 (IBM Corporation, New York, NY, USA). The Kolmogrov–Smirnov test was used for data normalization. The data were analyzed by one-way analysis of variance (ANOVA), and significant differences were determined using Duncan’s test (at a confidence level of *p* < 0.05). Measurements were conducted at least in triplicate and results were expressed as mean ± standard deviation.

## Figures and Tables

**Figure 1 gels-08-00021-f001:**
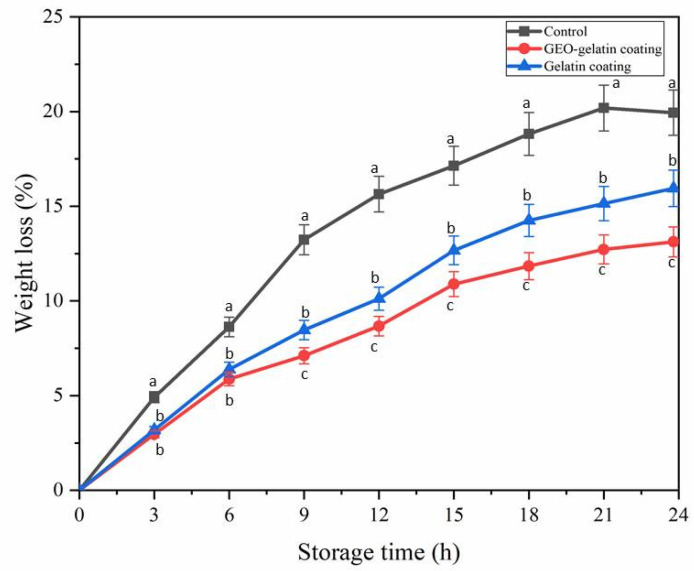
Weight loss rate of pork samples with different coating at 25 °C. Different letters in the same time indicate significant differences among samples (*p* < 0.05).

**Figure 2 gels-08-00021-f002:**
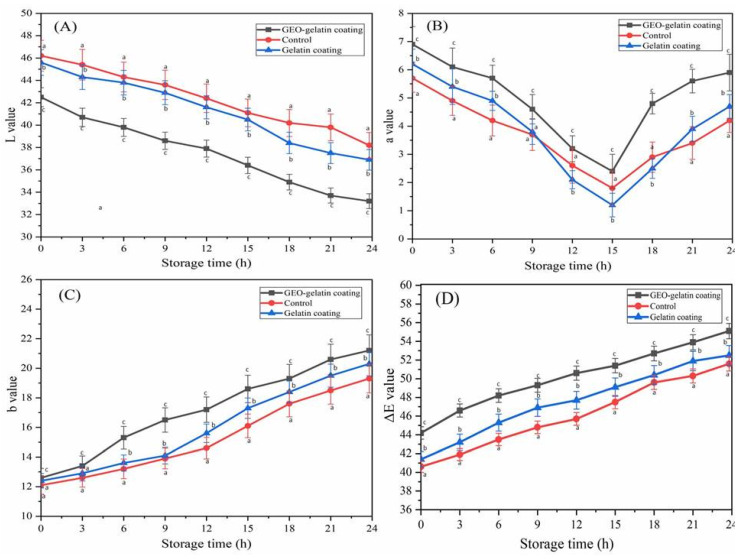
Color of pork with different coating at 25 °C, (**A**): *L* value; (**B**): *a** value; (**C**): *b** value; (**D**): Δ*E* value. Different letters in the same time indicate significant differences among samples (*p* < 0.05).

**Figure 3 gels-08-00021-f003:**
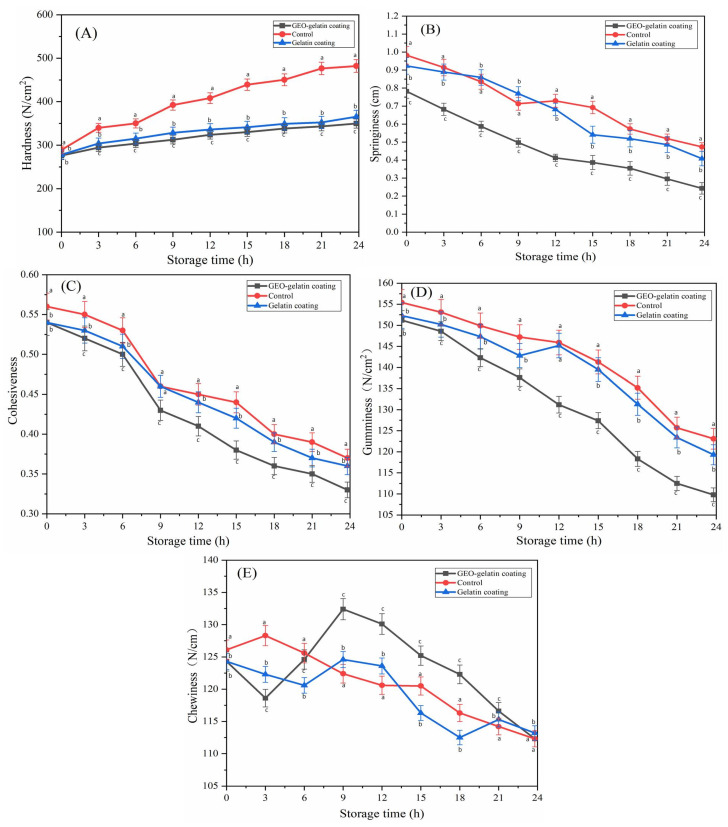
Texture of pork with different coating at 25 °C, (**A**): Hardness; (**B**): Springiness; (**C**): Cohesiveness; (**D**): Gumminess; (**E**): Chewiness. Different letters in the same time indicate significant differences among samples (*p* < 0.05).

**Figure 4 gels-08-00021-f004:**
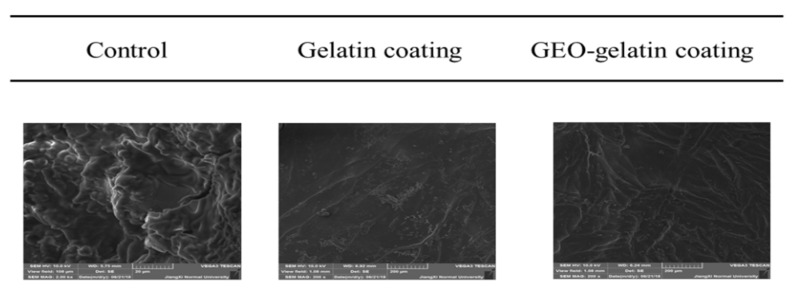
SEM micrographs of pork samples with different coating at room temperatures (25 °C).

**Figure 5 gels-08-00021-f005:**
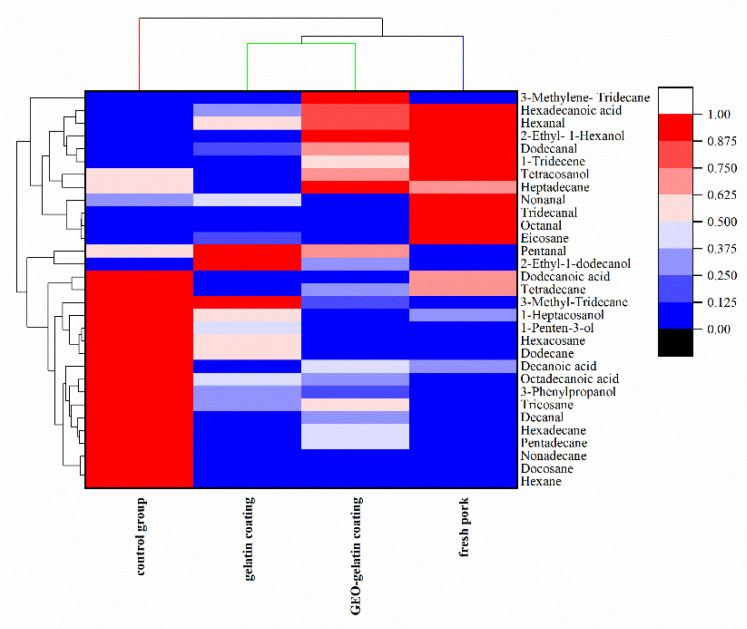
Heat map and cluster analysis of volatile components pork samples.

**Figure 6 gels-08-00021-f006:**
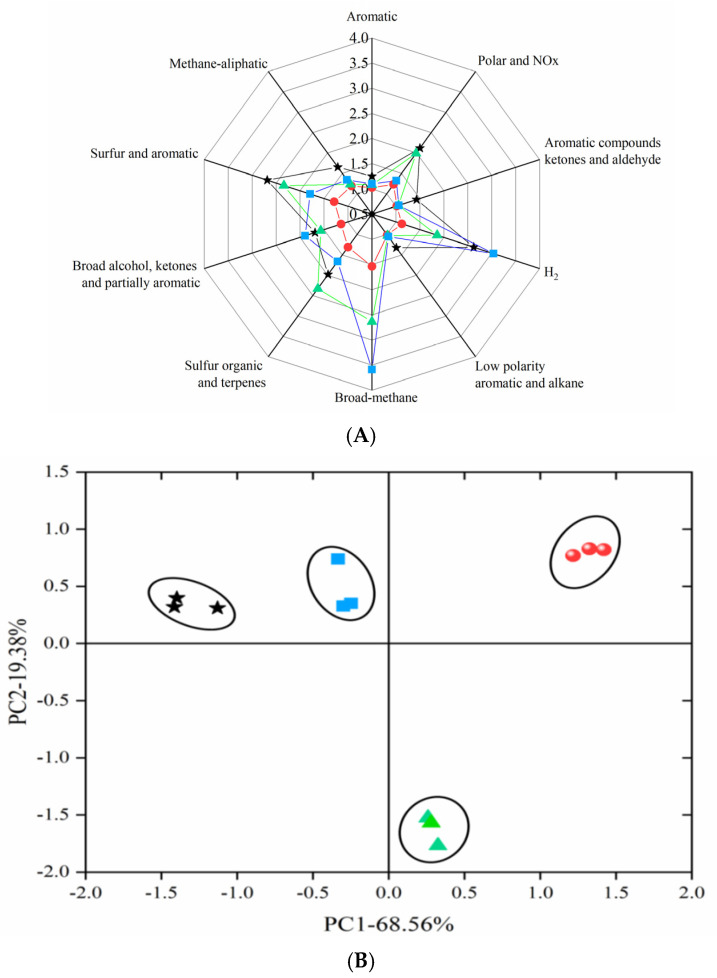
(**A**) E-nose of pork samples with different coating at 25 °C; (**B**). Principal component analysis (PCA) of E–nose. (

: control; 
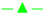
: gelatin coating; 
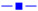
: GEO-gelatin coating; ★: fresh pork).

**Figure 7 gels-08-00021-f007:**
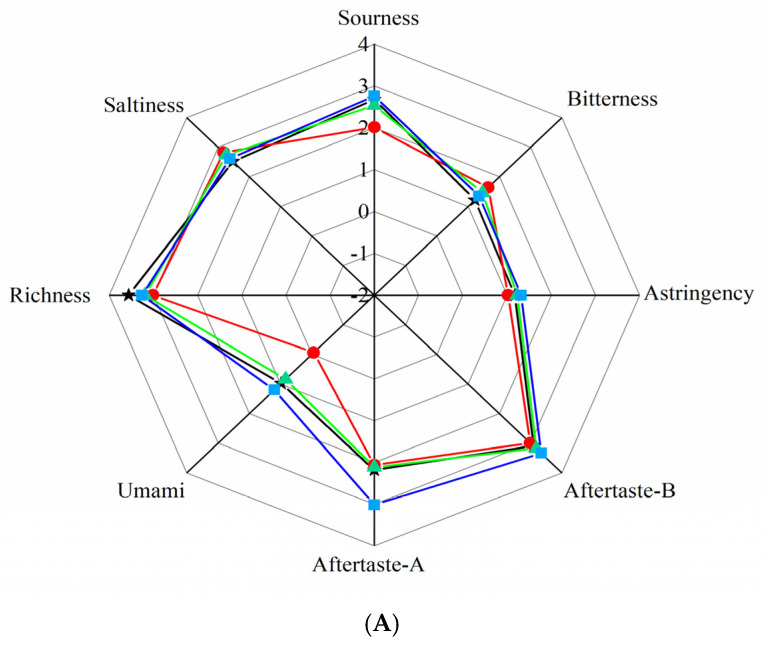
(**A**) E-tongue of pork samples with different coating at 25 °C; (**B**) Principal component analysis (PCA) of E-tongue. (

: control; 
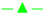
: gelatin coating; 
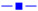
: GEO-gelatin coating; ★: fresh pork).

**Table 1 gels-08-00021-t001:** Volatile compounds (%) of different treatment of pork meat (means ± SD, *n* = 3). Mean values followed by different letters within a column are significantly different (*p* < 0.05).

Volatile Compound	Samples			
	Control Group	Gelatin Coating	GEO-Gelatin Coating	Fresh Pork
Alkane				
Hexane	0.31 ± 0.03 ^a^	N.D.	N.D.	N.D.
Dodecane	0.25 ± 0.03 ^a^	0.13 ± 0.01 ^b^	0.03 ± 0.04 ^c^	N.D.
Tetradecane	1.26 ± 0.10 ^a^	N.D.	0.34 ± 0.02 ^b^	0.83 ± 0.04 ^c^
3-Methyl-Tridecane	0.51 ± 0.04 ^a^	0.46 ± 0.04 ^a^	0.12 ± 0.02 ^b^	N.D.
Pentadecane	1.06 ± 0.08 ^a^	N.D.	0.42 ± 0.10 ^b^	N.D.
Eicosane	N.D.	0.39 ± 0.03 ^a^	N.D.	2.34 ± 0.10 ^b^
Docosane	1.03 ± 0.03 ^a^	N.D.	N.D.	N.D.
Hexadecane	0.65 ± 0.07 ^a^	0.04 ± 0.03 ^b^	0.32 ± 0.05 ^c^	N.D.
Heptadecane	0.18 ± 0.00 ^a^	N.D.	0.29 ± 0.03 ^b^	0.21 ± 0.03 ^a^
Tricosane	0.52 ± 0.09 ^a^	0.13 ± 0.02 ^b^	0.31 ± 0.05 ^c^	N.D.
Nonadecane	1.28 ± 0.06 ^a^	N.D.	0.12 ± 0.14 ^b^	N.D.
Hexacosane	0.78 ± 0.02 ^a^	0.43 ± 0.05 ^b^	N.D.	N.D.
alkene				
1-Tridecene	N.D.	N.D.	0.13 ± 0.06 ^a^	0.21 ± 0.02 ^b^
Tridecane, 3-methylene-	N.D.	N.D.	0.36 ± 0.01 ^a^	0.04 ± 0.04 ^b^
Alcohols				
2-Ethyl- Hexanol	0.19 ± 0.06 ^a^	0.14 ± 0.05 ^b^	3.59 ± 0.02 ^c^	3.63 ± 0.01 ^c^
3-Phenylpropanol	4.77 ± 0.03 ^a^	1.49 ± 0.01 ^b^	1.27 ± 0.02 ^c^	0.17 ± 0.05 ^d^
2-Ethyl-1-dodecanol	0.03 ± 0.09 ^a^	0.87 ± 0.01 ^b^	0.23 ± 0.06 ^c^	N.D.
1-Penten-3-ol	2.35 ± 0.01 ^a^	1.29 ± 0.03 ^b^	0.31 ± 0.01 ^c^	0.34 ± 0.09 ^c^
Tetracosanol	0.88 ± 0.03 ^a^	0.24 ± 0.10 ^b^	0.92 ± 0.15 ^a^	1.32 ± 0.03 ^c^
1-Heptacosanol	2.58 ± 0.06 ^a^	1.39 ± 0.02 ^b^	N.D.	0.81 ± 0.01 ^c^
Aldehyde				
Nonanal	1.30 ± 0.03 ^a^	1.74 ± 0.01 ^b^	N.D.	4.63 ± 0.16 ^c^
Decanal	0.14 ± 0.08 ^a^	0.06 ± 0.10 ^b^	0.08 ± 0.07 ^b^	0.05 ± 0.02 ^b^
Octanal	N.D.	0.16 ± 0.01 ^a^	0.08 ± 0.08 ^b^	1.39 ± 0.06 ^c^
Pentanal	2.65 ± 0.07 ^a^	5.13 ± 0.03 ^b^	3.42 ± 0.07 ^c^	N.D.
Hexanal	3.89 ± 0.02 ^a^	47.32 ± 2.03 ^b^	68.43 ± 0.08 ^c^	84.32 ± 0.05 ^d^
Dodecanal	N.D.	0.26 ± 0.07 ^a^	0.98 ± 0.09 ^b^	1.49 ± 0.03 ^c^
Tridecanal	0.15 ± 0.01 ^a^	0.09 ± 0.05 ^b^	N.D.	1.27 ± 0.02 ^c^
Acids				
Hexadecanoic acid	0.15 ± 0.03 ^a^	2.30 ± 0.07 ^b^	5.23 ± 0.08 ^c^	6.23 ± 0.05 ^d^
Dodecanoic acid	3.21 ± 0.12 ^a^	0.17 ± 0.05 ^b^	N.D.	2.13 ± 0.05 ^c^
Decanoic acid	0.32 ± 0.05 ^a^	N.D.	0.12 ± 0.04 ^b^	0.10 ± 0.09 ^b^
Octadecanoic acid	0.92 ± 0.10 ^a^	0.35 ± 0.02 ^b^	0.31 ± 0.05 ^b^	N.D.

N.D. not detected.

## Data Availability

The data presented in this study are available on request from the corresponding author.

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
