# Peer review of "Application of Gelatin Composite Coating in Pork Quality Preservation during Storage and Mechanism of Gelatin Composite Coating on Pork Flavor"

_gels, 2021, doi:10.3390/gels8010021_

Round 1
Reviewer 1 Report
Manuscript gels-1451220 reports on the Application of gelatin coating containing ginger essential oil on pork quality during storage at 25o C. This is a seemingly sound study with, however, numerous flaws and unclear description make the manuscript hard to understand especially when it comes to experimental design, lack of adequate discussion, non-conclusive results. Finally, the use of English is poor. My detailed comments follow the text sequence:
- The title does not match with material and methods. It is better to change the “film” in the title to “coating”.
- Abstract: should contain quantitative data, regarding the system studied.
- Lines 71-82. The statements made need a reference.
- It is necessary to add GC-MS analysis of essential oil.
- Line 99- The authors explain how they make the sterile environment
- Line 108- it is mentioned that the study period was 12 h. is it correct?
- How long did it take from slaughtering the pork to start the study?
- What kind of test was used in SPSS software?
- It is recommended to add significant differences in results of weight loss, color and etc. ( please add "P<0.05" at the end of the sentence)
- Cinnamaldehyde is one of the common compounds of cinnamon. How this compound is detected. If the GC-MS data is available, it can be concluded.
- The English grammar should be improved. As examples are mentioned:
- At first, dissolve gelatin (8%, based on deionized water) particles in deionized water, and then add glycerin (10%, w/w, based on the weight of gelatin). This is liquid A; Tween and GEO mixtures at the ratio of 1:1, this is liquid B; adding liquid B to liquid A so that the concentration of GEO in liquid A is 0.5% (ml/g, based on gelatin). For pure fish gelatin film, the GEO was removed. All the film-forming solutions were prepared and utilized simultaneously.
- the multisensory analysis of vision, touch, smell, and taste is used to study the effect of coating on preservation of pork flavor. By measuring the weight loss, color, texture parameters, micro structure,
Author Response
RESPONSES TO REVIEWER 1’S COMMENTS
(Please note that we first present your comments. We then offer our responses to each of your comments, using the bold format to do so.)
Manuscript gels-1451220 reports on the Application of gelatin coating containing ginger essential oil on pork quality during storage at 25℃. This is a seemingly sound study with, however, numerous flaws and unclear description make the manuscript hard to understand especially when it comes to experimental design, lack of adequate discussion, non-conclusive results. Finally, the use of English is poor. My detailed comments follow the text sequence:
Response: We truly appreciate your observations and value your supportive reaction to the nature of our work. We are grateful for the time and energy you put into developing this review. As we believe you will see, we have invested significant effort in preparing this revision to deal effectively with each of the issues you and the other reviewers identified in the initial version of our work. We are hopeful that you will agree that the revision is much improved and the theoretical and empirical contributions are much stronger.
‒ The title does not match with material and methods. It is better to change the “film” in the title to “coating”.
Response: Thank you for this valuable comment. “Coating” is indeed a much more appropriate expression. Thanks a lot. Thus in the revised manuscript, we revised the title as “Application of gelatin composite coating in pork quality preservation during storage and mechanism of gelatin composite coating on pork flavor”. Thank you again.
‒ Abstract: should contain quantitative data, regarding the system studied.
Response: Thanks for your suggestions. We have included the details of our quantitative data in abstract page on page 1. The revised abstract content is Aiming at solving the problem of flavor deterioration during the storage of pork at room temperature, pork coating technology was developed to preserve the pork at 25°C, and the multi-sensory analysis of vision, touch, smell, and taste was used to study the effect of coating on preservation of pork flavor. Uncoated(control)and coated pork samples were analyzed periodically during storage at 25°C for weight loss, color, texture (springiness, chewiness, cohesiveness, gumminess and hardness), micro structure, odor (electronic nose), taste(electronic tongue), volatile flavor substance and taste ingredients. The results suggested that ginger essential oil (GEO) gelatin coating and gelatin coating can effectively inhibit the loss of water dispersion and slow down the oxidation reaction, coating treatments could significantly (p < 0.05) retarded the weight loss of pork slices, with values of 20.19%, 15.95%, 13.12% for uncoated, gelatin coated and GEO-gelatin coated samples during 24 h of storage respectively. Compared with control group, the color, texture, smell and taste evaluations demonstrated that coating treatments had improved sensory and texture attributes during the storage period. . Furthermore, the comprehensive results from the physical property assays (especially the texture), morphological assay and volatile odor assays showed that the GEO-fish gelatin composite coating had better preservation effect on pork flavor than the fish gelatin coating. The antioxidant and antibacterial properties of ginger essential oil are better to inhibit the formation of lipid oxidation metabolites and microbial metabolites, so that the color, texture, smell and taste of pork can be better maintained. The study suggests that the gelatin composite coating could be developed as a prospective active packaging to preserve pork meat at room temperature.
‒ Lines 71-82. The statements made need a reference.
Response: Thank you for the detailed suggestion. We have rectified this error and carefully copy-edited the manuscript, added the related reference in this specific area (line73).
‒ It is necessary to add GC-MS analysis of essential oil.
Response: Thank you for your comments. Essential oil is a kind of volatile gas, and gelatin has a certain effect on inhibiting the volatilization of essential oil. In previous experiments, we found that when 0.5% ginger essential oil was added to the gelatin solution, the volatile components of ginger essential oil could not be detected by GC-MS, related paper has been published. In fact, we focused on flavor, antimicrobial activity and physical properties of gelatin films incorporated with quite low contents of ginger essential oil.
‒ Line 99- The authors explain how they make the sterile environment
Response: We are grateful for your suggestion. In the revised manuscript on page 3, we have illustrated the method how to make the sterile environment in details, " fifteen fresh pork boneless fillet were irradiated with a UV lamp on each side at a distance of 10–15 cm for 30 min prior to cutting into cuboids of dimensions 4 cm × 3 cm × 2 cm using a pre-sterilized knife and cutting board, subsequently the pork samples were prepared and placed in sterilized petri dishes. All reagents and operating apparatus in contact with pork samples must be sterilized at 121℃, under high pressure for 15-20 minutes in advance", thank you.
‒ Line 108- it is mentioned that the study period was 12 h. is it correct?
Response: Thank you for your question. Good catch, we apologize for the careless mistake. We have rectified this time period as during 24h of storage in line 125. Thank you.
‒ How long did it take from slaughtering the pork to start the study?
Response: Thank you for your suggestion. The fresh pork was purchased at 8 am from a market that near the lab in Guiyang (China) at 24 h post-mortem, after that we immediately carried out the preparation of pork samples and coating assays within 1 h. In addition, we have followed your suggestion by including a supplementary illustration in line 86 and 112. Thank you.
‒ What kind of test was used in SPSS software?
Response: Thank you for your question. SPSS® Statistics 25.0 (IBM Corporation, New York, USA) was used for statistical evaluations. The data were analyzed by one-way analysis of variance (ANOVA), and significant differences were determined using Duncan’s test (at a confidence level of P < 0.05). In the revised manuscript, we clarify the details of what kind of test used in SPSS software in line 196. Thanks for your valuable advice.
‒ It is recommended to add significant differences in results of weight loss, color and etc. ( please add "P<0.05" at the end of the sentence)
Response: Following your advice, we have added significant differences in results of weight loss, color, texture assays. In addition, at the end of the graph annotation, we added "Different letters in the same time indicate significant differences among samples (P < 0.05)".Thanks a lot for your valuable suggestion.
‒ Cinnamaldehyde is one of the common compounds of cinnamon. How this compound is detected. If the GC-MS data is available, it can be concluded.
Response: It should be a mistake that cinnamaldehyde exist in the original manuscript. This may be a residual ingredient in the instrument. It has been removed from the original manuscript. Thank you for your careful review. In addition, in order to minimize errors in data analysis. We completed the original data analysis of GC-MS again. Thus there are relatively major modifications in GC-MS part. We sincerely hope that works.
‒ The English grammar should be improved. As examples are mentioned:
At first, dissolve gelatin (8%, based on deionized water) particles in deionized water, and then add glycerin (10%, w/w, based on the weight of gelatin). This is liquid A; Tween and GEO mixtures at the ratio of 1:1, this is liquid B; adding liquid B to liquid A so that the concentration of GEO in liquid A is 0.5% (ml/g, based on gelatin). For pure fish gelatin film, the GEO was removed. All the film-forming solutions were prepared and utilized simultaneously.
The multi-sensory analysis of vision, touch, smell, and taste is used to study the effect of coating on preservation of pork flavor. By measuring the weight loss, color, texture parameters, micro structure,
Response: Thanks for your suggestion. In terms of the first example, we have revised as "at first, gelatin (8%, based on deionized water) particles were soaked and swelled in deionized water, and then glycerin (10%, w/w, based on the weight of gelatin) was added to the solution with continuous agitating for 10 min to obtain a uniform solution. This is liquid A; the content of liquid B was used to prepare mass ratio of Tween 80 and GEO mixtures 1:1. Active GEO-gelatin coating solution was prepared by adding liquid B to liquid A (GEO with a final concentration of 0.5% v/v). With respect to gelatin coating, this solution was prepared according to the preparation of GEO gelatin coating without adding GEO. All the film-forming solutions were prepared and utilized simultaneously". With respect to the second example, we’re sorry for the grammar mistakes and after careful consideration we have revised this part as "the multi-sensory analysis of vision, touch, smell, and taste is used to study the effect of coating on preservation of pork flavor. By measuring the weight loss, color, texture parameters, micro structure, volatile flavor substance and taste ingredients during storage, (it was found that ginger essential oil (GEO) gelatin coating and gelatin coating can effectively reduce the loss of water dispersion and slow down the oxidation reaction".
Furthermore, we have already checked the English throughout the manuscript by Elsevier English language service. However, we have further rectified the language as per your suggestions. Furthermore, we have polished the manuscript in details.
Overall response: We very much appreciate your constructive comments which were critical in guiding our revision. We hope you will agree that the revised manuscript is much improved both theoretically and empirically. Please know that we are more than happy to make any further additional changes if you deem necessary. Thanks for handling our manuscript and providing the opportunity for improving further. We have tried our best to work on every comment raised by the honorable reviewers to improve the manuscript. We have responded them point by point marked through track change options throughout the manuscript to trace them easily. Hopefully, now the manuscript would meet the requirements of “Gels” Journal.
Reviewer 2 Report
General comment
The manuscript Application of gelatin composite film in pork quality preserva- tion during storage and mechanism of gelatin composite coating on pork flavor showed the the active coating on pork .I did not identify a scientific novelty that would justify the article. The well-described state of the art could justify the research.
Specific comment
Metodologia
- Film Preparation is not clear
- 2.3 Pork Sample: what part of pork the slices were done? How many pork samples were used in the experments
- The study was performed 25 ℃ for 0, 3, 6, 9, 12, 15, 18, 21, 24 h, the fresh meat pork is stored under refrigeration temperature. An accelerated test at 25C not is an approach appropriated to demosntrate the efectivity of the active coating. The authors will add an assay at refrigeration temperature. Thus, will obtain solids results and discuss.
- Gas Chromatography-Mass Spectrometry: details are necessaries
- Resultas
- 20% of the weight loss shows that the coating was not efficient to retain the water.
- Color: The authors said that the initial a* value for all treatment increased and after decreased, but the Fig 2 (B) shows opposite behavior.
- The samples used to measured water loss and hardness is the same? By Fig 1the higher water loss occured for GEO-gelatin coating and the higher hardness it is observed for control, Please, clarify this.
- 3.2 line 320-323: "Furthermore, the pork meat of the coating film
presented a smooth surface owing to the coating film covers the pores on the surface of the pork meat, preventing the water from evaporating and the formation of wrinkles, which indicates that the coating has a protective effect on pork and prevents the wrinkling
performance of pork meat due to water loss during storage.". But the results showed opposite behavior. - Table 1: the results were obtained during the storage? When, which day? Which one affact negatively the pork meat?
Author Response
RESPONSES TO REVIEWER2’S COMMENTS
(Please note that we first present your comments. We then offer our responses to each of your comments, using the bold format to do so.)
Major comments:
The manuscript Application of gelatin composite film in pork quality preservation during storage and mechanism of gelatin composite coating on pork flavor showed the active coating on pork .I did not identify a scientific novelty that would justify the article. The well-described state of the art could justify the research.
Response: Thank you very much for your valuable to time to review and comment toward our manuscript. Thank you for your comments. We apologize for the errors in the last version of our work. We have carefully copy-edited the revised manuscript to minimize errors.
Metodologia
‒ Film Preparation is not clear
Response: We are grateful for your suggestion. In the revised manuscript on page 2, we have interpreted the film preparation in details, "At first, gelatin (8%, based on deionized water) particles were melted in deionized water, and then glycerin (10%, w/w, based on the weight of gelatin) was added to the solution with continuous agitating for 10 min to obtain a uniform solution. This is liquid A; the content of liquid B was used to prepare mass ratio of Tween 80 and GEO mixtures 1:1. Active GEO-gelatin coating solution was prepared by adding liquid B to liquid A (GEO with a final concentration of 0.5% v/v). With respect to gelatin coating, this solution was prepared according to the preparation of GEO gelatin coating without adding GEO. All the film-forming solutions were prepared and utilized simultaneously". Thank you.
‒ 2.3 Pork Sample: what part of pork the slices were done? How many pork samples were used in the experiments?
Response: Thank you for your comments. The slices were prepared from fresh pork fillet (derived from the griskin). Actually, the number of samples in each group was correlated with the type of test project. For example, in the TPA test, there are 8 parallel samples each time, on the other hand, in the color test, there are 6 parallel samples each time. Following your advice, we revised the materials and pork sample preparation part on page 2 and 3.
‒ The study was performed 25 ℃ for 0, 3, 6, 9, 12, 15, 18, 21, 24 h, the fresh meat pork is stored under refrigeration temperature. An accelerated test at 25 not is an approach appropriated to demonstrate the efficiency of the active coating. The authors will add an assay at refrigeration temperature. Thus, will obtain solids results and discuss.
Response: Thanks for your valuable suggestion. After careful consideration of the comments, we all argue that a single storage temperature is reasonable. Firstly, the main purpose of this experimental design is to compare the effects of the two coatings on the flavor of pork at room temperature, so as to obtain a more effective coating method and the mechanism of the coating's effect on the flavor of pork. Secondly, in fact, the sales locations are mainly supermarkets and farmer’s markets generally at room temperature without cold chain supply system. Fresh pork is often wasted due to a short shelf life during storage, simple plastic packaging is commonly used in the sales process, which leads to accelerated pork spoilage and deterioration of flavor and appearance. Thus it’s impossible to meet the consumers’ demand for high-quality pork. Considering that the meats tend to deteriorate at normal temperature compared to refrigeration temperature, hence our research focused on conducting assays at room temperature not refrigeration temperature. It’s worth mentioning that Dehnad et al. (2014) conducted the research referring to the effect of coating on meat under singer temperature(25℃). Besides, due to Covid spreading in China again, we are unable to get the testing again for these new conditions. Hopefully you will consider the actual situations.
‒ Gas Chromatography-Mass Spectrometry: details are necessaries
Response: Following your advice, we have added the details referring to GC-MS on pages 4. Thanks a lot for your valuable suggestion.
Results
‒ 20% of the weight loss shows that the coating was not efficient to retain the water.
Response: Thanks a lot for your comment, we’re sincerely sorry for the careless mistakes, as we labeled the wrong interpretation of each samples (including control, GEO-gelatin coating and gelatin coating). We have rectified this part in Fig 1. The weight loss of control group indicated the highest value (0% - 20.19%), higher than the weight loss of gelatin coating (0% - 15.95%) and GEO-gelatin coating (0% - 13.12%). The weight loss rate of the coated group was lower than that of the control group, and the effect of the GEO-gelatin group was slightly lower than that of the gelatin group. Therefore, the coating groups perform efficiently in retaining the water compared with the control group.
‒ Color: The authors said that the initial a* value for all treatment increased and after decreased, but the Fig 2 (B) shows opposite behavior.
Response: Thanks a lot for your comment, we’re sincerely sorry for the careless mistakes, as we clarified the statement in revised manuscript. We have rectified this part on page 7. We hope that works. Thanks a lot again, we really appreciate that.
‒ The samples used to measured water loss and hardness is the same? By Fig 1the higher water loss occurred for GEO-gelatin coating and the higher hardness it is observed for control, Please, clarify this.
Response: Thank you very much for your advice, we’re sincerely sorry for the careless mistakes, as we labeled the wrong interpretation of each samples (including control, GEO-gelatin coating and gelatin coating). We have rectified this part in Fig 1. We hope that works. Thanks a lot again, we really appreciate that.
‒ 3.2 line 320-323: "Furthermore, the pork meat of the coating film presented a smooth surface owing to the coating film covers the pores on the surface of the pork meat, preventing the water from evaporating and the formation of wrinkles, which indicates that the coating has a protective effect on pork and prevents the wrinkling performance of pork meat due to water loss during storage.". But the results showed opposite behavior.
Response: Thanks a lot for your comment, however, after careful consideration of the comments, we now argue that the results didn’t show the opposite behavior. As you can observe from the labeled circle, SEM images display the rough and wrinkled structure of control group, which causes the taste and flavor of the pork meat to deteriorate. While, pork surfaces are smooth in both coated groups, which indicates that the coating film plays a protective role in pork texture and prevents the phenomenon of pork wrinkling during storage due to water loss.
‒ Table 1: the results were obtained during the storage? When, which day? Which one affects negatively the pork meat?
Response: Thanks for your constructive questions. Actually, in this experiment, in order to explore the changes in the flavor of the pork during 24 h storage. As a result, it can be detected that the change of the flavor of pork with the extension of the pork storage time. From a microbiological point of view, meat products with a microbial contamination index of 7log CFU/g are deemed as stale meat. According to our previous experiments, the microbial contamination index of meat products kept at 25℃ for 18 hours amounted to 7 log CFU/g. Therefore, in the tests of GC-MS, electronic tongue, and electronic nose test, samples stored for 18 hours (control group, gelatin coating group and GEO coating group) were uniformly selected. During this time period (taking the control group as a reference), we compared the effects of two different coatings on pork flavor.
Overall Response: We are grateful for your guidance. We hope you will find the manuscript much improved. Please know that we are more than happy to make any further additional changes if you deem necessary. Thanks for handling our manuscript and providing the opportunity for improving further. We have tried our best to work on every comment raised by the honorable reviewers to improve the manuscript. We have responded them point by point marked through track change options throughout the manuscript to trace them easily. Hopefully, now the manuscript would meet the requirements of “Gels” Journal.

Reviewer 3 Report
Dear authors,
Please consider the suggestions below for improvement.
Abstract – The first three sentences should be rewritten to avoid confusions. The authors should mention with coatings were used before mention in the results. One suggestion for the first sentence: Gelatin coating incorporated with ginger essential oil was selected to investigate the integrity of pork during storage at room temperature.
Abstract (L17-17): confusing: which multi-sensory analysis are you talking about?
L31 – Microbiology analysis and antioxidant capacity assay were not made in this work, so the authors should mention how the coating affected the parameters studied in the manuscript
L40-41 – Instead of ‘’year by year’’ the authors should include the amount expected to be produced per year
L167-168 – Did not understand. When and how the internal standard was included?
L163-177 – Include in this topic the parameters found in GC-MS.
L201/L215/467/483 – Principle our principal component analysis? Standardize the nomenclature
L234 – Double check and fix the reference mentioned
L266-270 – Include reference that correlate color brightness with moisture content in meat. Compare your results with literature
L278 Kroll et al. (year).
L311-312 – Double check and fix the reference cited.
Figures 2 and 3 – Correct x axis: instead of ‘’Storage…hours’’ write: Storage time (h)
L347-348 – How the rough and wrinkled aspect is related to deterioration? The authors must explain
L 356 – Aksoy et al (year)
Reviewer 4 Report
This version has taken into account all the suggested revisions.
I suggest publication after a a final English language revision.
Best wishes!
Round 2
Reviewer 1 Report
The authors have now provided new data and some rebuttals to initial comments. The quality of the manuscript has improved but the question of GC-MS analysis of essential oil still remain.
Author Response
All the modifications are marked in red color in the text files
Reviewer 2 Report
The authors modiffied the manuscript.
Author Response

(The authors gave the same response as above.)
